# Pipeline PSRO: A Scalable Approach for Finding Approximate Nash Equilibria in Large Games

**Stephen McAleer**[*]
Department of Computer Science
University of California, Irvine
Irvine, CA
smcaleer@uci.edu

**John Lanier**[*]
Department of Computer Science
University of California, Irvine
Irvine, CA
jblanier@uci.edu

**Roy Fox**
Department of Computer Science
University of California, Irvine
Irvine, CA
royf@uci.edu

**Pierre Baldi**
Department of Computer Science
University of California, Irvine
Irvine, CA
pfbaldi@ics.uci.edu

## Abstract

Finding approximate Nash equilibria in zero-sum imperfect-information games is challenging when the number of information states is large. Policy Space Response Oracles (PSRO) is a deep reinforcement learning algorithm grounded in game theory that is guaranteed to converge to an approximate Nash equilibrium. However, PSRO requires training a reinforcement learning policy at each iteration, making it too slow for large games. We show through counterexamples and experiments that DCH and Rectified PSRO, two existing approaches to scaling up PSRO, fail to converge even in small games. We introduce Pipeline PSRO (P2SRO), the first scalable PSRO-based method for finding approximate Nash equilibria in large zero-sum imperfect-information games. P2SRO is able to parallelize PSRO with convergence guarantees by maintaining a hierarchical pipeline of reinforcement learning workers, each training against the policies generated by lower levels in the hierarchy. We show that unlike existing methods, P2SRO converges to an approximate Nash equilibrium, and does so faster as the number of parallel workers increases, across a variety of imperfect information games. We also introduce an open-source environment for Barrage Stratego, a variant of Stratego with an approximate game tree complexity of $10^{50}$. P2SRO is able to achieve state-of-the-art performance on Barrage Stratego and beats all existing bots. Experiment code is available at https://github.com/JBLanier/pipeline-psro.

## 1 Introduction

A long-standing goal in artificial intelligence and algorithmic game theory has been to develop a general algorithm which is capable of finding approximate Nash equilibria in large imperfect-information two-player zero-sum games. AlphaStar [Vinyals et al., 2019] and OpenAI Five [Berner et al., 2019] were able to demonstrate that variants of self-play reinforcement learning are capable of achieving expert-level performance in large imperfect-information video games. However, these methods are not principled from a game-theoretic point of view and are not guaranteed to converge to an approximate Nash equilibrium. Policy Space Response Oracles (PSRO) [Lanctot et al., 2017]

---

[*]Authors contributed equally

is a game-theoretic reinforcement learning algorithm based on the Double Oracle algorithm and is guaranteed to converge to an approximate Nash equilibrium.

PSRO is a general, principled method for finding approximate Nash equilibria, but it may not scale to large games because it is a sequential algorithm that uses reinforcement learning to train a full best response at every iteration. Two existing approaches parallelize PSRO: Deep Cognitive Hierarchies (DCH) [Lanctot et al., 2017] and Rectified PSRO [Balduzzi et al., 2019], but both have counterexamples on which they fail to converge to an approximate Nash equilibrium, and as we show in our experiments, neither reliably converges in random normal form games.

Although DCH approximates PSRO, it has two main limitations. First, DCH needs the same number of parallel workers as the number of best response iterations that PSRO takes. For large games, this requires a very large number of parallel reinforcement learning workers. This also requires guessing how many iterations the algorithm will need before training starts. Second, DCH keeps training policies even after they have plateaued. This introduces variance by allowing the best responses of early levels to change each iteration, causing a ripple effect of instability. We find that, in random normal form games, DCH rarely converges to an approximate Nash equilibrium even with a large number of parallel workers, unless their learning rate is carefully annealed.

Rectified PSRO is a variant of PSRO in which each learner only plays against other learners that it already beats. We prove by counterexample that Rectified PSRO is not guaranteed to converge to a Nash equilibrium. We also show that Rectified PSRO rarely converges in random normal form games.

In this paper we introduce Pipeline PSRO (P2SRO), the first scalable PSRO-based method for finding approximate Nash equilibria in large zero-sum imperfect-information games. P2SRO is able to scale up PSRO with convergence guarantees by maintaining a hierarchical pipeline of reinforcement learning workers, each training against the policies generated by lower levels in the hierarchy. P2SRO has two classes of policies: fixed and active. Active policies are trained in parallel while fixed policies are not trained anymore. Each parallel reinforcement learning worker trains an active policy in a hierarchical pipeline, training against the meta Nash equilibrium of both the fixed policies and the active policies on lower levels in the pipeline. Once the performance increase of the lowest-level active worker in the pipeline does not improve past a given threshold in a given amount of time, the policy becomes fixed, and a new active policy is added to the pipeline. P2SRO is guaranteed to converge to an approximate Nash equilibrium. Unlike Rectified PSRO and DCH, P2SRO converges to an approximate Nash equilibrium across a variety of imperfect information games such as Leduc poker and random normal form games.

We also introduce an open-source environment for Barrage Stratego, a variant of Stratego. Barrage Stratego is a large two-player zero sum imperfect information board game with an approximate game tree complexity of $10^{50}$. We demonstrate that P2SRO is able to achieve state-of-the-art performance on Barrage Stratego, beating all existing bots.

To summarize, in this paper we provide the following contributions:

- We develop a method for parallelizing PSRO which is guaranteed to converge to an approximate Nash equilibrium, and show that this method outperforms existing methods on random normal form games and Leduc poker.
- We present theory analyzing the performance of PSRO as well as a counterexample where Rectified PSRO does not converge to an approximate Nash equilibrium.
- We introduce an open-source environment for Stratego and Barrage Stratego, and demonstrate state-of-the-art performance of P2SRO on Barrage Stratego.

## 2   Background and Related Work

A two-player normal-form game is a tuple $(\Pi, U)$, where $\Pi = (\Pi_1, \Pi_2)$ is the set of policies (or strategies), one for each player, and $U : \Pi \to \mathbb{R}^2$ is a payoff table of utilities for each joint policy played by all players. For the game to be zero-sum, for any pair of policies $\pi \in \Pi$, the payoff $u_i(\pi)$ to player $i$ must be the negative of the payoff $u_{-i}(\pi)$ to the other player, denoted $-i$. Players try to maximize their own expected utility by sampling from a distribution over the policies $\sigma_i \in \Sigma_i = \Delta(\Pi_i)$. The set of best responses to a mixed policy $\sigma_i$ is defined as the set of policies

that maximally exploit the mixed policy: $\text{BR}(\sigma_i) = \arg\min_{\sigma'_{-i} \in \Sigma_{-i}} u_i(\sigma'_{-i}, \sigma_i)$, where $u_i(\sigma) = \text{E}_{\pi \sim \sigma}[u_i(\pi)]$. The exploitability of a pair of mixed policies $\sigma$ is defined as: $\text{EXPLOITABILITY}(\sigma) = \frac{1}{2}(u_2(\sigma_1, \text{BR}(\sigma_1)) + u_1(\text{BR}(\sigma_2), \sigma_2)) \geq 0$. A pair of mixed policies $\sigma = (\sigma_1, \sigma_2)$ is a Nash equilibrium if $\text{EXPLOITABILITY}(\sigma) = 0$. An approximate Nash equilibrium at a given level of precision $\epsilon$ is a pair of mixed policies $\sigma$ such that $\text{EXPLOITABILITY}(\sigma) \leq \epsilon$ [Shoham and Leyton-Brown, 2008].

In small normal-form games, Nash equilibria can be found via linear programming [Nisan et al., 2007]. However, this quickly becomes infeasible when the size of the game increases. In large normal-form games, no-regret algorithms such as fictitious play, replicator dynamics, and regret matching can asymptotically find approximate Nash equilibria [Fudenberg et al., 1998, Taylor and Jonker, 1978, Zinkevich et al., 2008]. Extensive form games extend normal-form games and allow for sequences of actions. Examples of perfect-information extensive form games include chess and Go, and examples of imperfect-information extensive form games include poker and Stratego.

In perfect information extensive-form games, algorithms based on minimax tree search have had success on games such as checkers, chess and Go [Silver et al., 2017]. Extensive-form fictitious play (XFP) [Heinrich et al., 2015] and counterfactual regret minimization (CFR) [Zinkevich et al., 2008] extend fictitious play and regret matching, respectively, to extensive form games. In large imperfect information games such as heads up no-limit Texas Hold 'em, counterfactual regret minimization has been used on an abstracted version of the game to beat top humans [Brown and Sandholm, 2018]. However, this is not a general method because finding abstractions requires expert domain knowledge and cannot be easily done for different games. For very large imperfect information games such as Barrage Stratego, it is not clear how to use abstractions and CFR. Deep CFR [Brown et al., 2019] is a general method that trains a neural network on a buffer of counterfactual values. However, Deep CFR uses external sampling, which may be impractical for games with a large branching factor such as Stratego and Barrage Stratego. DREAM [Steinberger et al., 2020] and ARMAC [Gruslys et al., 2020] are model-free regret-based deep learning approaches. Current Barrage Stratego bots are based on imperfect information tree search and are unable to beat even intermediate-level human players [Schadd and Winands, 2009, Jug and Schadd, 2009].

Recently, deep reinforcement learning has proven effective on high-dimensional sequential decision making problems such as Atari games and robotics [Li, 2017]. AlphaStar [Vinyals et al., 2019] beat top humans at Starcraft using self-play and population-based reinforcement learning. Similarly, OpenAI Five [Berner et al., 2019] beat top humans at Dota using self play reinforcement learning. Similar population-based methods have achieved human-level performance on Capture the Flag [Jaderberg et al., 2019]. However, these algorithms are not guaranteed to converge to an approximate Nash equilibrium. Neural Fictitious Self Play (NFSP) [Heinrich and Silver, 2016] approximates extensive-form fictitious play by progressively training a best response against an average of all past policies using reinforcement learning. The average policy is represented by a neural network and is trained via supervised learning using a replay buffer of past best response actions. This replay buffer may become prohibitively large in complex games.

## 2.1 Policy Space Response Oracles

The Double Oracle algorithm [McMahan et al., 2003] is an algorithm for finding a Nash equilibrium in normal form games. The algorithm works by keeping a population of policies $\Pi^t \subset \Pi$ at time $t$. Each iteration a Nash equilibrium $\sigma^{*,t}$ is computed for the game restricted to policies in $\Pi^t$. Then, a best response to this Nash equilibrium for each player $\text{BR}(\sigma^{*,t}_{-i})$ is computed and added to the population $\Pi^{t+1}_i = \Pi^t_i \cup \{\text{BR}(\sigma^{*,t}_{-i})\}$ for $i \in \{1, 2\}$.

Policy Space Response Oracles (PSRO) approximates the Double Oracle algorithm. The meta Nash equilibrium is computed on the empirical game matrix $U^\Pi$, given by having each policy in the population $\Pi$ play each other policy and tracking average utility in a payoff matrix. In each iteration, an approximate best response to the current meta Nash equilibrium over the policies is computed via any reinforcement learning algorithm. In this work we use a discrete-action version of Soft Actor Critic (SAC), described in Section 3.1.

One issue with PSRO is that it is based on a normal-form algorithm, and the number of pure strategies in a normal form representation of an extensive-form game is exponential in the number of information sets. In practice, however, PSRO is able to achieve good performance in large games,

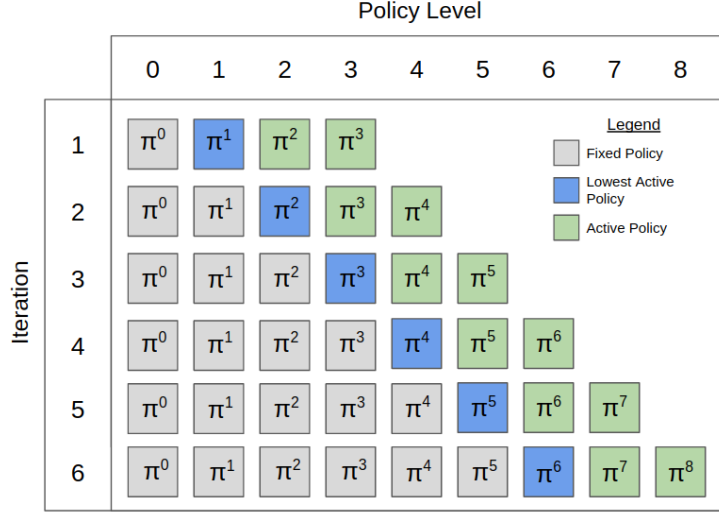

Figure 1: Pipeline PSRO. The lowest-level active policy $\pi^j$ (blue) plays against the meta Nash equilibrium $\sigma^{*,j}$ of the lower-level fixed policies in $\Pi^f$ (gray). Each additional active policy (green) plays against the meta Nash equilibrium of the fixed and training policies in levels below it. Once the lowest active policy plateaus, it becomes fixed, a new active policy is added, and the next active policy becomes the lowest active policy. In the first iteration, the fixed population consists of a single random policy.

possibly because large sections of the game tree correspond to weak actions, so only a subset of pure strategies need be enumerated for satisfactory performance. Another issue with PSRO is that it is a sequential algorithm, requiring a full best response computation in every iteration. This paper addresses the latter problem by parallelizing PSRO while maintaining the same convergence guarantees.

DCH [Lanctot et al., 2017] parallelizes PSRO by training multiple reinforcement learning agents, each against the meta Nash equilibrium of agents below it in the hierarchy. A problem with DCH is that one needs to set the number of workers equal to the number of policies in the final population beforehand. For large games such as Barrage Stratego, this might require hundreds of parallel workers. Also, in practice, DCH fails to converge in small random normal form games even with an exact best-response oracle and a learning rate of 1, because early levels may change their best response occasionally due to randomness in estimation of the meta Nash equilibrium. In our experiments and in the DCH experiments in Lanctot et al. [2017], DCH is unable to achieve low exploitability on Leduc poker.

Another existing parallel PSRO algorithm is Rectified PSRO [Balduzzi et al., 2019]. Rectified PSRO assigns each learner to play against the policies that it currently beats. However, we prove that Rectified PSRO does not converge to a Nash equilibrium in all symmetric zero-sum games. In our experiments, Rectified PSRO rarely converges to an approximate Nash equilibrium in random normal form games.

## 3 Pipeline Policy Space Response Oracles (P2SRO)

Pipeline PSRO (P2SRO; Algorithm 1) is able to scale up PSRO with convergence guarantees by maintaining a hierarchical pipeline of reinforcement learning policies, each training against the policies in the lower levels of the hierarchy (Figure 1). P2SRO has two classes of policies: fixed and active. The set of fixed policies are denoted by $\Pi^f$ and do not train anymore, but remain in the fixed population. The parallel reinforcement learning workers train the active policies, denoted $\Pi^a$ in a hierarchical pipeline, training against the meta Nash equilibrium distribution of both the fixed policies and the active policies in levels below them in the pipeline. The entire population $\Pi$ consists of the union of $\Pi^f$ and $\Pi^a$. For each policy $\pi_i^j$ in the active policies $\Pi_i^a$, to compute the distribution

---

**Algorithm 1** Pipeline Policy-Space Response Oracles

---

**Input:** Initial policy sets for all players $\Pi^f$
Compute expected utilities for empirical payoff matrix $U^\Pi$ for each joint $\pi \in \Pi$
Compute meta-Nash equilibrium $\sigma^{*,j}$ over fixed policies ($\Pi^f$)
**for** many episodes **do**
    **for all** $\pi^j \in \Pi^a$ in parallel **do**
        **for** player $i \in \{1, 2\}$ **do**
            Sample $\pi_{-i} \sim \sigma_{-i}^{*,j}$
            Train $\pi_i^j$ against $\pi_{-i}$
        **end for**
        **if** $\pi^j$ plateaus and $\pi^j$ is the lowest active policy **then**
            $\Pi^f = \Pi^f \cup \{\pi^j\}$
            Initialize new active policy at a higher level than all existing active policies
            Compute missing entries in $U^\Pi$ from $\Pi$
            Compute meta Nash equilibrium for each active policy
        **end if**
        Periodically compute meta Nash equilibrium for each active policy
    **end for**
**end for**
Output current meta Nash equilibrium on whole population $\sigma^*$

---

of policies to train against, a meta Nash equilibrium $\sigma_{-i}^{*,j}$ is periodically computed on policies lower than $\pi_i^j$: $\Pi_{-i}^f \cup \{\pi_{-i}^k \in \Pi_{-i}^a | k < j\}$ and $\pi_i^j$ trains against this distribution.

The performance of a policy $\pi^j$ is given by the average performance during training $\mathbb{E}_{\pi_1 \sim \sigma_1^{*,j}}[u_2(\pi_1, \pi_2^j)] + \mathbb{E}_{\pi_2 \sim \sigma_2^{*,j}}[u_1(\pi_1^j, \pi_2)]$ against the meta Nash equilibrium distribution $\sigma^{*,j}$. Once the performance of the lowest-level active policy $\pi^j$ in the pipeline does not improve past a given threshold in a given amount of time, we say that the policy's performance plateaus, and $\pi^j$ becomes fixed and is added to the fixed population $\Pi^f$. Once $\pi^j$ is added to the fixed population $\Pi^f$, then $\pi^{j+1}$ becomes the new lowest active policy. A new policy is initialized and added as the highest-level policy in the active policies $\Pi^a$. Because the lowest-level policy only trains against the previous fixed policies $\Pi^f$, P2SRO maintains the same convergence guarantees as PSRO. Unlike PSRO, however, each policy in the pipeline above the lowest-level policy is able to get a head start by pre-training against the moving target of the meta Nash equilibrium of the policies below it. Unlike Rectified PSRO and DCH, P2SRO converges to an approximate Nash equilibrium across a variety of imperfect information games such as Leduc Poker and random normal form games.

In our experiments we model the non-symmetric games of Leduc poker and Barrage Stratego as symmetric games by training one policy that can observe which player it is at the start of the game and play as either the first or the second player. We find that in practice it is more efficient to only train one population than to train two different populations, especially in larger games, such as Barrage Stratego.

### 3.1 Implementation Details

For the meta Nash equilibrium solver we use fictitious play [Fudenberg et al., 1998]. Fictitious play is a simple method for finding an approximate Nash equilibrium in normal form games. Every iteration, a best response to the average strategy of the population is added to the population. The average strategy converges to an approximate Nash equilibrium. For the approximate best response oracle, we use a discrete version of Soft Actor Critic (SAC) [Haarnoja et al., 2018, Christodoulou, 2019]. We modify the version used in RLlib [Liang et al., 2018, Moritz et al., 2018] to account for discrete actions.

### 3.2 Analysis

PSRO is guaranteed to converge to an approximate Nash equilibrium and doesn't need a large replay buffer, unlike NFSP and Deep CFR. In the worst case, all policies in the original game must be added

before PSRO reaches an approximate Nash equilibrium. Empirically, on random normal form games, PSRO performs better than selecting pure strategies at random without replacement. This implies that in each iteration, PSRO is more likely than random to add a pure strategy that is part of the support of the Nash equilibrium of the full game, suggesting the conjecture that PSRO has faster convergence rate than random strategy selection. The following theorem indirectly supports this conjecture.

**Theorem 3.1.** *Let $\sigma$ be a Nash equilibrium of a symmetric normal form game $(\Pi, U)$ and let $\Pi^e$ be the set of pure strategies in its support. Let $\Pi' \subset \Pi$ be a population that does not cover $\Pi^e \not\subseteq \Pi'$, and let $\sigma'$ be the meta Nash equilibrium of the original game restricted to strategies in $\Pi'$. Then there exists a pure strategy $\pi \in \Pi^e \setminus \Pi'$ such that $\pi$ does not lose to $\sigma'$.*

*Proof.* Contained in supplementary material. $\square$

Ideally, PSRO would be able to add a member of $\Pi^e \setminus \Pi'$ to the current population $\Pi'$ at each iteration. However, the best response to the current meta Nash equilibrium $\sigma'$ is generally not a member of $\Pi^e$. Theorem 3.1 shows that for an *approximate* best response algorithm with a weaker guarantee of not losing to $\sigma'$, it is possible that a member of $\Pi^e \setminus \Pi'$ is added at each iteration.

Even assuming that a policy in the Nash equilibrium support is added at each iteration, the convergence of PSRO to an approximate Nash equilibrium can be slow because each policy is trained sequentially by a reinforcement learning algorithm. DCH, Rectified PSRO, and P2SRO are methods of speeding up PSRO through parallelization. In large games, many of the basic skills (such as extracting features from the board) may need to be relearned when starting each iteration from scratch. DCH and P2SRO are able to speed up PSRO by pre-training each level on the moving target of the meta Nash equilibrium of lower-level policies before those policies converge. This speedup would be linear with the number of parallel workers if each policy could train on the fixed final meta Nash equilibrium of the policies below it. Since it trains instead on a moving target, we expect the speedup to be sub-linear in the number of workers.

DCH is an approximation of PSRO that is not guaranteed to converge to an approximate Nash equilibrium if the number of levels is not equal to the number of pure strategies in the game, and is in fact guaranteed *not* to converge to an approximate Nash equilibrium if the number of levels cannot support it.

Another parallel PSRO algorithm, Rectified PSRO, is not guaranteed to converge to an approximate Nash equilibrium.

**Proposition 3.1.** *Rectified PSRO with an oracle best response does not converge to a Nash equilibrium in all symmetric two-player, zero-sum normal form games.*

*Proof.* Consider the following symmetric two-player zero-sum normal form game:

$$\begin{bmatrix} 0 & -1 & 1 & -\frac{2}{5} \\ 1 & 0 & -1 & -\frac{2}{5} \\ -1 & 1 & 0 & -\frac{2}{5} \\ \frac{2}{5} & \frac{2}{5} & \frac{2}{5} & 0 \end{bmatrix}$$

This game is based on Rock–Paper–Scissors, with an extra strategy added that beats all other strategies and is the pure Nash equilibrium of the game. Suppose the population of Rectified PSRO starts as the pure Rock strategy.

- Iteration 1: Rock ties with itself, so a best response to Rock (Paper) is added to the population.

- Iteration 2: The meta Nash equilibrium over Rock and Paper has all mass on Paper. The new strategy that gets added is the best response to Paper (Scissors).

- Iteration 3: The meta Nash equilibrium over Rock, Paper, and Scissors equally weights each of them. Now, for each of the three strategies, Rectified PSRO adds a best response to the meta-Nash-weighted combination of strategies that it beats or ties. Since Rock beats or ties Rock and Scissors, a best response to a $50 - 50$ combination of Rock and Scissors is Rock,

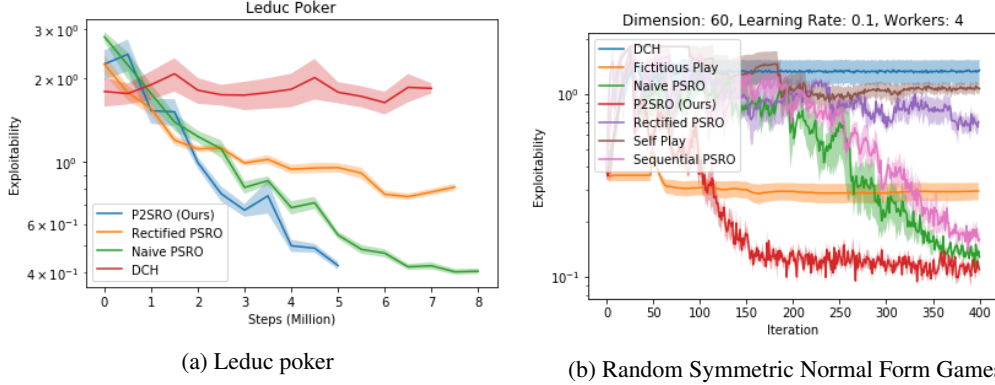

(a) Leduc poker

(b) Random Symmetric Normal Form Games

Figure 2: Exploitability of Algorithms on Leduc poker and Random Symmetric Normal Form Games

with an expected utility of $\frac{1}{2}$. Similarly, for Paper, since Paper beats or ties Paper and Rock, a best response to a $50-50$ combination of Paper and Rock is Paper. For Scissors, the best response for an equal mix of Scissors and Paper is Scissors. So in this iteration no strategy is added to the population and the algorithm terminates.

We see that the algorithm terminates without expanding the fourth strategy. The meta Nash equilibrium of the first three strategies that Rectified PSRO finds are not a Nash equilibrium of the full game, and are exploited by the fourth strategy, which is guaranteed to get a utility of $\frac{2}{5}$ against any mixture of them. □

The pattern of the counterexample presented here is possible to occur in large games, which suggests that Rectified PSRO may not be an effective algorithm for finding an approximate Nash equilibrium in large games. Prior work has found that Rectified PSRO does not converge to an approximate Nash equilibrium in Kuhn Poker [Muller et al., 2020].

**Proposition 3.2.** *P2SRO with an oracle best response converges to a Nash equilibrium in all two-player, zero-sum normal form games.*

*Proof.* Since only the lowest active policy can be submitted to the fixed policies, this policy is an oracle best response to the meta Nash distribution of the fixed policies, making P2SRO with an oracle best response equivalent to the Double Oracle algorithm. □

Unlike DCH which becomes unstable when early levels change, P2SRO is able to avoid this problem because early levels become fixed once they plateau. While DCH only approximates PSRO, P2SRO has equivalent guarantees to PSRO because the lowest active policy always trains against a fixed meta Nash equilibrium before plateauing and becoming fixed itself. This fixed meta Nash distribution that it trains against is in principle the same as the one that PSRO would train against. The only difference between P2SRO and PSRO is that the extra workers in P2SRO are able to get a head-start by pre-training on lower level policies while those are still training. Therefore, P2SRO inherits the convergence guarantees from PSRO while scaling up when multiple processors are available.

## 4  Results

We compare P2SRO with DCH, Rectified PSRO, and a naive way of parallelizing PSRO that we term Naive PSRO. Naive PSRO is a way of parallelizing PSRO where each additional worker trains against the same meta Nash equilibrium of the fixed policies. Naive PSRO is beneficial when randomness in the reinforcement learning algorithm leads to a diversity of trained policies, and in our experiments it performs only slightly better than PSRO. Additionally, in random normal form game experiments, we include the original, non-parallel PSRO algorithm, termed sequential PSRO, and non-parallelized self-play, where a single policy trains against the latest policy in the population.

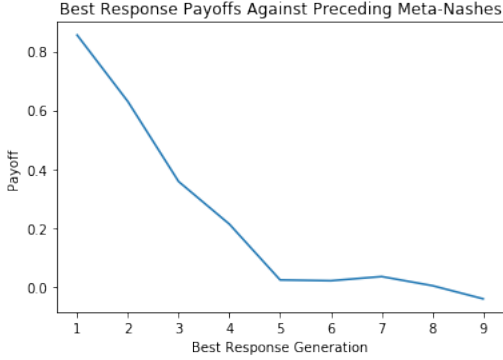

Figure 3: Barrage Best Response Payoffs Over Time

| Name | P2SRO Win Rate vs. Bot |
|---|---|
| Asmodeus | 81% |
| Celsius | 70% |
| Vixen | 69% |
| Celsius1.1 | 65% |
| **All Bots Average** | **71%** |

Table 1: Barrage P2SRO Results vs. Existing Bots

We find that DCH fails to reliably converge to an approximate Nash equilibrium across random symmetric normal form games and small poker games. We believe this is because early levels can randomly change even after they have plateaued, causing instability in higher levels. In our experiments, we analyze the behavior of DCH with a learning rate of 1 in random normal form games. We hypothesized that DCH with a learning rate of 1 would be equivalent to the double oracle algorithm and converge to an approximate Nash. However, we found that the best response to a fixed set of lower levels can be different in each iteration due to randomness in calculating a meta Nash equilibrium. This causes a ripple effect of instability through the higher levels. We find that DCH almost never converges to an approximate Nash equilibrium in random normal form games.

Although not introduced in the original paper, we find that DCH converges to an approximate Nash equilibrium with an annealed learning rate. An annealed learning rate allows early levels to not continually change, so the variance of all of the levels can tend to zero. Reinforcement learning algorithms have been found to empirically converge to approximate Nash equilibria with annealed learning rates [Srinivasan et al., 2018, Bowling and Veloso, 2002]. We find that DCH with an annealed learning rate does converge to an approximate Nash equilibrium, but it can converge slowly depending on the rate of annealing. Furthermore, annealing the learning rate can be difficult to tune with deep reinforcement learning, and can slow down training considerably.

## 4.1 Random Symmetric Normal Form Games

For each experiment, we generate a random symmetric zero-sum normal form game of dimension $n$ by generating a random antisymmetric matrix $P$. Each element in the upper triangle is distributed uniformly: $\forall i < j \leq n$, $a_{i,j} \sim \text{UNIFORM}(-1, 1)$. Every element in the lower triangle is set to be the negative of its diagonal counterpart: $\forall j < i \leq n$, $a_{i,j} = -a_{j,i}$. The diagonal elements are equal to zero: $a_{i,i} = 0$. The matrix defines the utility of two pure strategies to the row player. A strategy $\pi \in \Delta^n$ is a distribution over the $n$ pure strategies of the game given by the rows (or equivalently, columns) of the matrix. In these experiments we can easily compute an exact best response to a strategy and do not use reinforcement learning to update each strategy. Instead, as a strategy $\pi$ "trains" against another strategy $\hat{\pi}$, it is updated by a learning rate $r$ multiplied by the best response to that strategy: $\pi' = r\text{BR}(\hat{\pi}) + (1 - r)\pi$.

Figure 2 show results for each algorithm on random symmetric normal form games of dimension 60, about the same dimension of the normal form of Kuhn poker. We run each algorithm on five different random symmetric normal form games. We report the mean exploitability over time of these algorithms and add error bars corresponding to the standard error of the mean. P2SRO reaches an approximate Nash equilibrium much faster than the other algorithms. Additional experiments on different dimension games and different learning rates are included in the supplementary material. In each experiment, P2SRO converges to an approximate Nash equilibrium much faster than the other algorithms.

### 4.2 Leduc Poker

Leduc poker is played with a deck of six cards of two suits with three cards each. Each player bets one chip as an ante, then each player is dealt one card. After, there is a a betting round and then another card is dealt face up, followed by a second betting round. If a player's card is the same rank as the public card, they win. Otherwise, the player whose card has the higher rank wins. We run the following parallel PSRO algorithms on Leduc: P2SRO, DCH, Rectified PSRO, and Naive PSRO. We run each algorithm for three random seeds with three workers each. Results are shown in Figure 2. We find that P2SRO is much faster than the other algorithms, reaching 0.4 exploitability almost twice as soon as Naive PSRO. DCH and Rectified PSRO never reach a low exploitability.

### 4.3 Barrage Stratego

Barrage Stratego is a smaller variant of the board game Stratego that is played competitively by humans. The board consists of a ten-by-ten grid with two two-by-two barriers in the middle. Initially, each player only knows the identity of their own eight pieces. At the beginning of the game, each player is allowed to place these pieces anywhere on the first four rows closest to them. More details about the game are included in the supplementary material.

We find that the approximate exploitability of the meta-Nash equilibrium of the population decreases over time as measured by the performance of each new best response. This is shown in Figure 3, where the payoff is 1 for winning and -1 for losing. We compare to all existing bots that are able to play Barrage Stratego. These bots include: Vixen, Asmodeus, and Celsius. Other bots such as Probe and Master of the Flag exist, but can only play Stratego and not Barrage Stratego. We show results of P2SRO against the bots in Table 1. We find that P2SRO is able to beat these existing bots by $71\%$ on average after $820,000$ episodes, and has a win rate of over $65\%$ against each bot. We introduce an open-source environment for Stratego, Barrage Stratego, and smaller Stratego games at `https://github.com/JBLanier/stratego_env`.

## Broader Impact

Stratego and Barrage Stratego are very large imperfect information board games played by many around the world. Although variants of self-play reinforcement learning have achieved grandmaster level performance on video games, it is unclear if these algorithms could work on Barrage Stratego or Stratego because they are not principled and fail on smaller games. We believe that P2SRO will be able to achieve increasingly good performance on Barrage Stratego and Stratego as more time and compute are added to the algorithm. We are currently training P2SRO on Barrage Stratego and we hope that the research community will also take interest in beating top humans at these games as a challenge and inspiration for artificial intelligence research.

This research focuses on how to scale up algorithms for computing approximate Nash equilibria in large games. These methods are very compute-intensive when applied to large games. Naturally, this favors large tech companies or governments with enough resources to apply this method for large, complex domains, including in real-life scenarios such as stock trading and e-commerce. It is hard to predict who might be put at an advantage or disadvantage as a result of this research, and it could be argued that powerful entities would gain by reducing their exploitability. However, the same players already do and will continue to benefit from information and computation gaps by exploiting suboptimal behavior of disadvantaged parties. It is our belief that, in the long run, preventing exploitability and striving as much as practical towards a provably efficient equilibrium can serve to level the field, protect the disadvantaged, and promote equity and fairness.

## Acknowledgments and Disclosure of Funding

SM and PB in part supported by grant NSF 1839429 to PB.

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
