[Supplementary Material]

# Supplementary Materials For
# Pipeline PSRO: A Scalable Approach for Finding Approximate Nash Equilibria in Large Games

## A Proofs of Theorems

**Theorem A.1.** *Let $\sigma$ be a Nash equilibrium of a symmetric normal form game $(\Pi, U)$ and let $\Pi^e$ be the set of pure strategies in its support. Let $\Pi' \subset \Pi$ be a population that does not cover $\Pi^e \not\subseteq \Pi'$, and let $\sigma'$ be the meta Nash equilibrium of the original game restricted to strategies in $\Pi'$. Then there exists a pure strategy $\pi \in \Pi^e \setminus \Pi'$ such that $\pi$ does not lose to $\sigma'$.*

*Proof.* $\sigma'$ is a meta Nash equilibrium, implying $\sigma'^{\intercal} G \sigma' = 0$, where $G$ is the payoff matrix for the row player. In fact, each policy $\pi$ in the support $\Pi'^e$ of $\sigma'$ has $1_\pi^{\intercal} G \sigma' = 0$, where $1_\pi$ is the one-hot encoding of $\pi$ in $\Pi$.

Consider the sets $\Pi^+ = \{\pi : \sigma(\pi) > \sigma'(\pi)\} = \Pi^e \setminus \Pi'$ and $\Pi^- = \{\pi : \sigma(\pi) < \sigma'(\pi)\} \subseteq \Pi'^e$. Note the assumption that $\Pi^+$ is not empty. If each $\pi \in \Pi^+$ had $1_\pi^{\intercal} G \sigma' < 0$, we would have

$$\sigma^{\intercal} G \sigma' = (\sigma - \sigma')^{\intercal} G \sigma' = \sum_{\pi \in \Pi^+} (\sigma(\pi) - \sigma'(\pi)) 1_\pi^{\intercal} G \sigma' + \sum_{\pi \in \Pi^-} (\sigma(\pi) - \sigma'(\pi)) 1_\pi^{\intercal} G \sigma'$$
$$= \sum_{\pi \in \Pi^+} (\sigma(\pi) - \sigma'(\pi)) 1_\pi^{\intercal} G \sigma' < 0,$$

in contradiction to $\sigma$ being a Nash equilibrium. We conclude that there must exist $\pi \in \Pi^+$ with $1_\pi^{\intercal} G \sigma' \geq 0$. $\qquad\square$

## B Barrage Stratego Details

Barrage is a smaller variant of the board game Stratego that is played competitively by humans. The board consists of a ten-by-ten grid with two two-by-two barriers in the middle (see image for details). Each player has eight pieces, consisting of one Marshal, one General, one Miner, two Scouts, one Spy, one Bomb, and one Flag. Crucially, each player only knows the identity of their own pieces. At the beginning of the game, each player is allowed to place these pieces anywhere on the first four rows closest to them.

The Marshal, General, Spy, and Miner may move only one step to any adjacent space but not diagonally. Bomb and Flag pieces cannot be moved. The Scout may move in a straight line like a rook in chess. A player can attack by moving a piece onto a square occupied by an opposing piece. Both players then reveal their piece's rank and the weaker piece gets removed. If the pieces are of equal rank then both get removed. The Marshal has higher rank than all other pieces, the General has higher rank than all other beside the Marshal, the Miner has higher rank than the Scout, Spy, Flag, and Bomb, the Scout has higher rank than the Spy and Flag, and the Spy has higher rank than the Flag and the Marshal when it attacks the Marshal. Bombs cannot attack but when another piece besides the Miner attacks a Bomb, the Bomb has higher rank. The player who captures his/her opponent's Flag or prevents the other player from moving any piece wins.

Figure 1: Valid Barrage Stratego Setup (note that the piece values are not visible to the other player)

## C  Additional Random Normal Form Games Results

We compare the exploitability over time of P2SRO with DCH, Naive PSRO, Rectified PSRO, Self Play, and Sequential PSRO. We run 5 experiments for each set of dimension, learning rate, and number of parallel workers and record the average exploitability over time. We run experiments on dimensions of size 15, 30, 45, 60, and 120, learning rates of 0.1, 0.2, and 0.5, and 4, 8, and 16 parallel workers. We find that not only does P2SRO converge to an approximate Nash equilibrium in every experiment, but that it performs as good as or better than all other algorithms in every experiment. We also find that the relative performance of P2SRO versus the other algorithms seems to improve as the dimension of the game improves.

Dimension: 60, Learning Rate: 0.1, Workers: 4

Dimension: 60, Learning Rate: 0.1, Workers: 8

Dimension: 60, Learning Rate: 0.1, Workers: 16

Dimension: 60, Learning Rate: 0.2, Workers: 4

Dimension: 60, Learning Rate: 0.2, Workers: 8

Dimension: 60, Learning Rate: 0.2, Workers: 16

Dimension: 60, Learning Rate: 0.5, Workers: 4

Dimension: 60, Learning Rate: 0.5, Workers: 8

Dimension: 60, Learning Rate: 0.5, Workers: 16

Dimension: 45, Learning Rate: 0.1, Workers: 4

Dimension: 45, Learning Rate: 0.1, Workers: 8

Dimension: 45, Learning Rate: 0.1, Workers: 16

Dimension: 45, Learning Rate: 0.2, Workers: 4

Dimension: 45, Learning Rate: 0.2, Workers: 8

Dimension: 45, Learning Rate: 0.2, Workers: 16

Dimension: 45, Learning Rate: 0.5, Workers: 4

Dimension: 45, Learning Rate: 0.5, Workers: 8

Dimension: 45, Learning Rate: 0.5, Workers: 16