[Reviews · NeurIPS 2020]

Review 1

Summary and Contributions: The authors introduce a novel algorithm to find approximate Nash equilibria in very large extensive form games called Pipeline PSRO (P2PSRO). The algorithm is an extension of PSRO algorithm introduce by Lanctot et. al. The authors indentified both computational and convergence weaknesses in the original PSRO alglrithm and provided elegant fixes.

Strengths: The paper proposes one important fix to the original PSRO algorithm, which simultaneously increases its algorithmic complexity and improves its convergence properties: rather than keeping a fixed number of cognitive hierarchies and continuously training all of the in parallel, P2PSRO algorithm stops training policies at a given level once they stop improving. Workers that become free can start training other policies at the next level. This allows to solve the following problems: 1. The number of cognitive hierarchies no longer has to be known in advance. 2. As all but one policies from which the approximate Nash is constructed are no longer changing, this improves stability properties. 3. Stopping training policies once they no longer improve can significantly improve computational complexity of the algorithm. The authors demonstrate that their algorithm can outperform PSRO and Rectified PSRO both in theory and in practice. They trained an agent on Stratego Barrage and demonstrated that the resulting policy was winning against several popular Stratego bots.

Weaknesses: The main problem which I have is about evaluation. I am not sure if it is enough to test any algorithm attempting to find a Nash equilibria against a fixed set of opponent bots, as this does not demonstrate how exploitable the policy actually is, as opponent bots are fixed and do not adapt to exploit the policy. It would be better to keep the final P2PSRO policy fixed and train an RL algorithm to play against it. If an exploit was found, this means that the policy was exmploitable. My biggest worry about any version of PSRO playing on very large extensive form games is the following. Imagine that PSRO is producing an approximate Nash by mixing N deterministic policies. Imagine that a game has M steps. Every time when an opponent observes a stochastic move, it can rule out the PSRO bot playing a bunch of its deterministic component policies, until exactly one policy is left. Then that policy can be exploited. Thus, in order to produce an actual Nash equilibria, PSRO is likely to need an exponential number of pure policies in the number of moves in the game in its mixture. For this reason I would suspect that P2PSRO policies could actually be highly exploitable in large games (like Barrage) while this would not be revealed by playing against a fixed number of opponent bots. One of the cheapest ways to get an estimate of exploitability is to see by how much the last hierarchy is beating the current Nash. Ideally, it would be nice to train a fixed policy with a _larger_ network against the final PSRO policy for a very long time and see by how much P2PSRO policy could be exploited.

Correctness: I could not find any obvious flaws.

Clarity: The paper is very well written and was a pure pleasure to read.

Relation to Prior Work: Prior work is properly acknowledged.

Reproducibility: Yes

Additional Feedback:


Review 2

Summary and Contributions: This paper shows that previous scalable/distributed variants of PSRO are not guaranteed to converge. It then follows to introduce a sound scalable variant - PipelinePSRO (P2SRO). The experimental section includes results on Leduc and Barrage Stratego.

Strengths: The paper is well motivated, and showing counterexamples to the previous PSRO variants is a valuable result. I liked the example/proof of proposition 3.1. The paper is well explained.

Weaknesses: The paper is missing a comparison with the most relevant previous work, namely XFP [1] Heinrich, Johannes, and David Silver. "Deep reinforcement learning from self-play in imperfect-information games." and DeepCFR [2] Brown, Noam, et al. "Deep counterfactual regret minimization." Both of these works are mentioned in the Background and Related Work, but: 1) XFP is just mentioned but never compared to in experiments 2)DeepCFR is just discarded with “However, Deep CFR uses external sampling, which may be impractical for games with a large branching factor such as Stratego and Barrage Stratego.” - if that’s the case, it should be supported by some evidence. Furthermore, there are newer variants based on this work, and it is not limited to a particular form of sampling. The paper only really compares to other variants from the PSRO family Furthermore, the theory and algorithms (the way described in the text) deal only with matrix games, while the experiments are on extensive form games. If the goal is to run on top of the exponentially large matrix game, this should be discussed. The theory included in the paper is rather trivial. Unfortunately, the experimental section does not make up for it. Leduc Results: The resulting numbers are far from impressive, exploitability of 0.4 in Leduc is just not good. Both [1] and [2] report better results in this game. Stratego Results: “We compare to all existing bots that are able to play Barrage Stratego” seems to be too strong statement as the comparison is only against a (subset) of agents in a particular github repository (https://github.com/braathwaate/strategoevaluator). The evaluation is also missing arguably the strongest agent mentioned - Probe (“the three time Computer Stratego World Champion” https://probe.soft112.com/) as the text mentions that it can not play Barrage, but the Probe’s homepage states that it does support the barrage variant.

Correctness: All seems correct.

Clarity: The paper is written relatively well.

Relation to Prior Work: Authors state that “We will release our code in an open-source repository.” but do not include the repository link at this time.

Reproducibility: Yes

Additional Feedback: Abstract: “We introduce Pipeline PSRO (P2SRO), the first scalable general method for finding approximate Nash equilibria in large zero-sum imperfect-information games.” -> Maybe use “the first scalable PSRO-based method “ as there are quite a few previous general and scalable methods outside of the PSRO family. Proposition 3.2. Talks about matrix games. As your experiments and domain in question is mostly about extensive form games, I should add some discussion regarding extensive form games. ***************************************************************************************************************************************************************** Post Rebuttal: I am happy with the authors response regarding the Barrage Stratego evaluation. I still think that this paper is not particularly strong as the theory is trivial and the comparison to previous work is more or less lacking, but this is probably enough.


Review 3

Summary and Contributions: This paper proposes a principled approach to fix issues of DCH and Rectified PSRO. It proposes an algorithm that converges to a Nash equilibrium and is able to scale to the game of Stratego Barrage.

Strengths: This paper makes a very nice contribution to the field of multiagent reinforement learning. - The counter-example to DCH and Rectified PSRO are very sound, - The proposed algorithm is definitely fixing these issues and seem to scale to Nash equilibrium in the large game of Stratego Barrage.

Weaknesses: The limitations of the proposed method is the length of the best response. But this drawback is typical of algorithms based on best responses.

Correctness: The method seem correct as far as I can tell.

Clarity: yes.

Relation to Prior Work: the relation to prior is discussed at length.

Reproducibility: Yes

Additional Feedback:


Review 4

Summary and Contributions: This article presents a parallel extension of the PSRO algorithm which is used to find approximate Nash equilibria in zero-sum games. In contrast to previous extensions to PSRO, the authors propose an algorithm that works well with bounded compute and that retains convergence guarantees.

Strengths: The article is very well presented, and provides better extensions to PSRO than existing methods in the literature. The analysis on random normal form games, where the Nash can be explicitly calculated provides a sound comparison against other techniques. The results on Poker and Stratego further the validity and broaden the generality claim of this new algorithm.

Weaknesses: This work is relatively incremental, in that, the main contribution is to assign a fixed number of active learners to computing best responses to previous strategies in the PSRO hierarchical progression. The compute is bounded by fixing policies that have plateaued, thus releasing a learner to start computing the next hierarchical best response. That said, at least a few other articles exist in the literature that do something similar, with worse guarantees of performance, so this is obviously not a terrible shortcoming.

Correctness: The article proposes a theoretically sound way to extend PSRO. Importantly, the extension is particularly amenable to use available compute, regardless of the size of the original problem. This makes future users of this algorithm have a clear tradeoff between speedup from parallelisation against the cost of compute. Previous extensions to PSRO were not as clear, where compute would have a potentially disproportionate effect on final performance.

Clarity: Yes, the article is well presented and the language is clear and at the right level of abstraction and detail.

Relation to Prior Work: Yes, the article is well contextualised

Reproducibility: Yes

Additional Feedback: lines 80-83: It would be useful to have a citation on this paragraph. In particular, people familiar with Nash equilibria, and nor PSRO, might wonder how Exploitability, which assumes two players evaluating changing strategies, relates to Nash, where only a single player evaluates a change of strategy. The choice for Barrage Stratego is unclear. Perhaps a more explicit reasoning would be desirable. line 107: Typo: "Similaraly"

[Author Response · NeurIPS 2020]

We would like to sincerely thank each of the reviewers for their time and for their insightful comments.

**Reviewer 1** You are right that with any version of PSRO, a potentially exponential number of pure strategies is needed
for an extensive form game. However, despite this limitation, PSRO is a very popular and promising algorithm, and
successful algorithms such as AlphaStar are based on it. You are correct that we do not actually find an approximate
Nash equilibrium in Barrage Stratego. It is our belief that Pipeline PSRO with enough compute would lead to expert-
level play in Stratego and Barrage Stratego, although it might still be subtly exploitable due to the reason you mention.
Also, as you mentioned, this limitation is due to PSRO itself, and not our improvement of PSRO, which performs better
than PSRO, DCH, and Rectified PSRO. We will add further analysis on the exploitability of the algorithm on Barrage.
We will include both of your suggestions: training a policy from scratch against the final meta-Nash, and analyzing
the final performance of each best response during training. We have analyzed the performance of the best response
over time against the meta-Nash in Barrage Stratego and we see that the performance goes down over time, providing
evidence that the meta-Nash is becoming less exploitable during training.

**Reviewer 2** We did not have sufficient computing power to compare with Deep CFR or NFSP on Barrage Stratego
or Leduc. However, we would expect both of these algorithms to outperform any PSRO variant (including Pipeline
PSRO) on Leduc poker based on the results reported in these papers. As for Barrage Stratego or similar large games,
it is somewhat of an open question whether NFSP or Deep CFR would outperform PSRO-based algorithms. One
of the downsides with NFSP is the need to store a large replay buffer of all past experience. Since we are fairly
limited with storage, the amount of storage needed to store enough experience to get good results on Barrage Stratego
could be too large. Similarly, it is unclear if Deep CFR would be able to get good results on Barrage Stratego. As
mentioned in the Deep CFR paper (and in private conversations with the author), the large branching factor could
be a problem, but we are not sure if it would or not. After we submitted this paper, DREAM was posted to arxiv:
https://arxiv.org/pdf/2006.10410.pdf which only samples a single action at each decision point. We think that this
method would be more promising for Barrage Stratego than NFSP or Deep CFR. We would be very interested to see
results of NFSP, Deep CFR, and DREAM on Barrage Stratego, but these experiments were outside the scope of this
work.

The theory behind PSRO works in both normal form and extensive form games, but you are correct that in extensive-form
games, PSRO could require an exponential number of pure strategies. We will edit the paper to clearly describe this
drawback of PSRO. Despite not performing as well as NFSP and Deep CFR on Leduc, PSRO is still a very promising
approach for large games. AlphaStar, which was inspired by PSRO, achieved expert-level performance on StarCraft,
and it is not clear that a method like NFSP or DREAM would have achieved a similar level. For these reasons, work on
PSRO is a promising and important research direction. Existing approaches to parallelizing PSRO are unstable (DCH)
or not guaranteed to converge (Rectified PSRO). Among PSRO variants, Pipeline PSRO is clearly state-of-the-art based
on theoretical guarantees and empirical performance. Our work is an important contribution to getting PSRO to work at
scale.

To summarize: PSRO does not perform as well as Deep CFR and NFSP on Leduc poker and could require an exponential
number of pure strategies in extensive form games. Despite these drawbacks, PSRO remains a popular and promising
algorithm, and it is an open question whether PSRO would perform better than these algorithms on large games.
Existing approaches to parallelizing PSRO are unstable or are not guaranteed to converge. Pipeline PSRO is the first
approach to parallelize PSRO while maintaining PSRO's convergence guarantees.

Other points: We have been in contact with the author of Probe in order to compare with it. During these discussions,
we asked if Probe played Barrage, to which we were explicitly told "no". The software description at the link provided
by the reviewer appears to conflict with this, however all download links that we could find to this software (including
the reviewer's) are dead. We are in the process of resolving this with the author of Probe, as a Barrage comparison
should be done if it is possible. All other bots that we found such as Master of the Flag were not able to play Barrage.
We will rephrase the abstract to describe P2SRO as "the first scalable PSRO-based method". We will add discussion
about how in extensive form games, a potentially exponential number of pure strategies are required to guarantee
convergence to a Nash equilibrium. We have added normal form experiments comparing to fictitious play, which gives
somewhat of a comparison to NFSP on normal form games and we find that PSRO outperforms fictitious play.

**Reviewer 3** You are correct that a limitation to PSRO is that it requires a long best response. Our approach is able to
significantly shorten this by training each best response in a hierarchy, but it still has this drawback.

**Reviewer 4** You are right that Pipeline PSRO is a simple change to PSRO. We think that this simplicity is a strength
because it is very easy to replicate and understand. Furthermore, we show drastically improved performance compared
with DCH and Rectified PSRO, which are similar to our algorithm. We will add a citation to lines 80-83 and will clear
up the language. We will add more reasoning on why we chose Barrage Stratego. The main reason is that it is a larger
game and has many more turns per game than heads-up Texas hold 'em, which only has four turns. We will fix the typo.

[Meta-Review · NeurIPS 2020]

The paper presents an improvement to policy-space response oracles (PSRO) algorithm, in particular its parallel form DCH. The fix is straight-forward, but the effect it has in practice is important as demonstrated by the authors across known benchmarks and randomized games. Please take the reviews into account when preparing the final version. If possible, it might be useful to include a mention of Probe Stratego/Barrage agent, and a summary of the reasons that it could not be used as a benchmark at this time.